# Tomato Chlorosis Virus Infection Facilitates *Bemisia tabaci* MED Reproduction by Elevating *Vitellogenin* Expression

**DOI:** 10.3390/insects12020101

**Published:** 2021-01-25

**Authors:** Liping Huang, Xiaobin Shi, Jizhe Shi, Zhuo Zhang, Yong Fang, Zhanhong Zhang, Qiuyi Pan, Limin Zheng, Yang Gao, Deyong Zhang, Xinqiu Tan, Yong Liu, Xuguo Zhou

**Affiliations:** 1Longping Branch Graduate School, Hunan University, Changsha 410125, China; hlp1936580786@hnu.edu.cn (L.H.); shixiaobin@hunaas.cn (X.S.); deyongzhang@hunaas.cn (D.Z.); 2Institute of Plant Protection, Hunan Academy of Agricultural Sciences, Changsha 410125, China; zhangzhuo@hunaas.cn (Z.Z.); zhenglimin@hunaas.cn (L.Z.); gaoyang@hunaas.cn (Y.G.); tanxinqiu@hunaas.cn (X.T.); 3Department of Entomology, University of Kentucky, Lexington, KY 40546, USA; jizhe.shi@uky.edu; 4Institute of Biotechnology, Hunan Academy of Agricultural Sciences, Changsha 410125, China; yongfang@hunaas.cn; 5Institute of Vegetable, Hunan Academy of Agricultural Sciences, Changsha 410125, China; zhangzhanhong@hunaas.cn; 6Plant Protection and Plant Inspection Station, Changde 415000, China; anxiangzhibao@hunaas.cn

**Keywords:** *Bemisia tabaci*, fecundity, ovarian development, gene expression, RNA interference, *vitellogenin*

## Abstract

**Simple Summary:**

The sweet potato whitefly, *Bemisia tabaci*, is a polyphagous, global invasive insect pest. It can damage vegetables and crops directly by feeding and indirectly by transmitting plant viruses. Previously, we showed that virus infection of host plants can promote *B. tabaci* MED (Q biotype) reproduction. Here, using a whitefly-tomato chlorosis virus (ToCV)-tomato system, we investigated how ToCV modulates *B. tabaci* reproduction to facilitate its spread. ToCV infection significantly increased whitefly fecundity and the relative expression of vitellogenin gene (*Vg*). Both ovarian development and fecundity of whitefly were suppressed when *Vg* expression was silenced with or without ToCV infection. These combined results reveal that ToCV infection increases *B. tabaci* MED fecundity via elevated vitellogenin gene expression.

**Abstract:**

Transmission of plant pathogenic viruses mostly relies on insect vectors. Plant virus could enhance its transmission by modulating the vector. Previously, we showed that feeding on virus infected plants can promote the reproduction of the sweet potato whitefly, *Bemisia tabaci* MED (Q biotype). In this study, using a whitefly-Tomato chlorosis virus (ToCV)-tomato system, we investigated how ToCV modulates *B. tabaci* MED reproduction to facilitate its spread. Here, we hypothesized that ToCV-infected tomato plants would increase *B. tabaci* MED fecundity via elevated *vitellogenin (Vg)* gene expression. As a result, fecundity and the relative expression of *B. tabaci* MED *Vg* was measured on ToCV-infected and uninfected tomato plants on days 4, 8, 12, 16, 20 and 24. The role of *Vg* on *B. tabaci* MED reproduction was examined in the presence and absence of ToCV using dietary RNAi. ToCV infection significantly increased *B. tabaci* MED fecundity on days 12, 16 and 20, and elevated *Vg* expression on days 8, 12 and 16. Both ovarian development and fecundity of *B. tabaci* MED were suppressed when *Vg* was silenced with or without ToCV infection. These combined results suggest that ToCV infection increases *B. tabaci* MED fecundity via elevated *Vg* expression.

## 1. Introduction

Plant viruses are a serious threat to sustainable agriculture [1,2,3]. The transmission of plant pathogenic viruses mostly relies on insect vectors to the shared host plants [4,5,6]. The complex interaction between the plant pathogenic viruses, the insect vectors, and the host plants could either directly or indirectly influence the abundance and distribution of the insect vectors [7], thereby affecting the spread of such virus diseases. For example, viruses could directly influence the insect vectors’ reproductive potential, impacting the abundance of the insect vectors and then affecting the transmission efficiency of the viruses or indirectly through the shared host plants modification. However, mechanisms, especially at the molecular level, governing vector behaviors are lacking [8].

The sweet potato whitefly, *Bemisia tabaci* (Gennadius) (Hemiptera: Aleyrodidae), is a destructive insect pest causing millions to billions dollars of damages in agriculture around the globe [9]. *Bemisia tabaci* could damage crops directly by feeding, or indirectly by excreting “honeydew” as a substrate for fungus growing, or by transmitting viruses [10]. Middle East Asia Minor 1 (MEAM1, previously known as the B biotype) and Mediterranean (MED, previously known as the Q biotype) are the two most devastating and invasive *B. tabaci* cryptic species worldwide. There are more than 200 plant viruses transmitted by *B. tabaci* causing severe yield reduction [11,12,13]. *B. tabaci* and its associated plant viruses have caused substantial crop losses ranging from 20 to 100% [14] with increasing control costs [15]. Among these plant viruses, tomato chlorosis virus (ToCV, genus *Crinivirus*, family Closteroviridae) is a globally distributed, rapidly-spread virus [16]. ToCV is semi-persistently transmitted by *B. tabaci* with incidences of up to 100% on tomato plants [17,18]. Because of a higher virus acquisition and accumulation rate and a better performance (e.g., fecundity) on ToCV infected tomato plants than other cryptic species, *B. tabaci* MED has been suggested to account for the rapid spread of ToCV [16]. Since no commercial ToCV-resistant cultivar has been identified in tomato now [19], control of the whitefly vector is critical in ToCV control in tomato. Deciphering the mechanism of ToCV mediating whitefly reproduction could help understand ToCV epidemiology and develop novel and effective control methods.

Such interaction has been suggested in a whitefly-virus-tomato system, in which *B. tabaci* MED fecundity was increased when feeding on ToCV-infected tomato plants [16]. Similarly, when infected with ToCV, *vitellogenin* (*Vg*) was the most differentially expressed gene in *B. tabaci* MEAM1 [20]. *Vitellogenin*, a precursor of the major yolk protein in most oviparous species and invertebrates, is synthesized mainly by the fat body, then secreted into the hemolymph and absorbed by the growing oocytes through receptor-mediated endocytosis, and finally served as the nutrients for developing embryos [21,22,23]. Such differential expression suggests *Vg* is a key factor in ToCV mediated whitefly reproduction, however, the mechanism is not clear.

Our goal here is to investigate how ToCV modulates whitefly reproduction. In this study, we hypothesized that ToCV-infected tomato plants would increase whitefly fecundity via elevated *vitellogenin* expression. To test this hypothesis, we (1) documented *B. tabaci* MED fecundity on ToCV-infected and uninfected tomato plants on days 4, 8, 12, 16, 20 and 24; (2) measured the relative expression of *B. tabaci* MED *vitellogenin* on ToCV-infected and uninfected tomato plants on days 4, 8, 12, 16, 20 and 24; and finally, (3) examined the role of *vitellogenin* on *B. tabaci* MED reproduction in the presence and absence of ToCV using dietary RNAi.

## 2. Materials and Methods

### 2.1. Host Plants and Whitefly Rearing

Tomato plants (*Solanum lycopersicum,* Zuanhongmeina, Institute of Vegetable, Hunan Academy of Agricultural Sciences) were kept in whitefly-proof screen-cages in a greenhouse at L16: D8, 25 ± 1 °C, and R.H. 60 ± 10%. To generate ToCV infected tomato plants, 0.5 mL ToCV infectious cDNA agro clone was injected into the three-true-leaf stage tomato plants. Both visual (leaf chlorosis) and molecular (RT-PCR) inspections were conducted to confirm the viral infection [9].

*Bemisia tabaci* MED was collected originally from infested poinsettia (*Euphorbia pulcherrima*) in Beijing, China in 2009, and maintained on tomato plants in whitefly-proof screen-cages at L14: D10, 25 ± 1 °C, and 60 ± 10% R.H [24].The purity of *Bemisia tabaci* MED population was monitored every month based on the *mitochondrial cytochrome oxidase I* (*mtCOI*) gene [25,26].

### 2.2. Whitefly Fecundity on ToCV-Infected and Uninfected Tomato Plants

Newly enclosed female whiteflies were collected and transferred onto either ToCV-infected or uninfected tomato plants. Each whitefly was contained in one clip-cage (Figure 1A) attached to one leaf (Figure 1B). There were five clip-cages on each plant (Figure 1C, attached to the second to fifth leaf from each branch top). A Leica M205C stereomicroscope (Wetzlar, Germany) was used to determine the sex of whitefly by observing the tail tip of the whiteflies (Figure 1D). The number of eggs in each clip-cage were counted under the stereomicroscope every 4 days for 24 days. The whitefly with its clip-cage was then transferred to a new plant after each count. Infected whiteflies were kept being transferred to the infected tomato plant, whereas the uninfected ones were transferred to the uninfected plants as control. The mortality of whiteflies was monitored every day after the first transfer. Thirty replicated clip-cages were made for infected and uninfected tomato plants, respectively.

### 2.3. Whitefly Vg Expression on ToCV-Infected and Uninfected Tomato Plants

Approximately 500 newly emerged whitefly adults were placed on ToCV-infected and uninfected tomato plants, respectively. We found the developmental period for *B. tabaci* was 15 to 20 days in tomato plants when incubated in the whitefly rearing room. Whiteflies were transferred to new ToCV-infected or uninfected tomato plants every 12 days to avoid the newly eclosed *B. tabaci* influence. Whiteflies with a series of feeding time (4, 8, 12, 16, 20, 24 days) were collected as samples, with 30 individuals in each and there were 3 biological replicates.

Total RNA was extracted from each sample by LIFE 15596-026 TRIzol^®^ Reagent (Life Technologies, Beijing, China). The concentration and purity of RNA were measured by NanoDrop 2000 (Thermo Fisher Scientific, Beijing, China) and 200 ng of which was used to synthesize the first strand of cDNA for qRT-PCR by using TransScript All-in-One First-Strand cDNA synthesis SuperMix for qPCR (One-Step gDNA Removal) kit (TransGen Biotech, Beijing, China) in accordance with the manufacturer’s instructions. Corresponding qPCR primers were designed according to the *Vg* sequence, and qRT-PCR was carried out on qTOWER3G qPCR system (Analytik Jena, Jena, Germany) by using TransStart Green qPCR SuperMix UDG kit (TransGen Biotech, Beijing, China). The first strand of cDNA was used as the template, and the “qPCR *Vg*-F” and “qPCR *Vg*-R” in Table 1 were used as specific primers. All reactions were performed in three technical replicates. Normalized gene expression was calculated using the 2^−ΔΔCT^ method [27] with *Actin* and *EF-1α* as the reference gene (Table 1) [16,28]. There were 9 replicates in each treatment, with 3 biological replicates and 3 technical replicates.

### 2.4. The Role of Vg on Whitefly Reproduction in the Presence and Absence of ToCV

#### 2.4.1. dsRNA Synthesis

The *Vg* and *GFP* gene sequences were amplified by Phanta^®^ Max Super-Fidelity DNA Polymerase kit (Vazyme Biotech, Nanjing, China). The cDNA of *B. tabaci* as the template, ds*Vg*-F/ds*Vg*-R and ds*GFP*-F/ds*GFP*-R containing the T7 RNA polymerase promoter sequence at the 5′ end as specific primers (Table 1). The PCR amplification procedure was as follows: 95 °C for 3 min; 35 cycles of 95 °C for 15 s, 60 °C for 15 s, 72 °C for 1 min; and 72 °C for 5 min. PCR products were detected by 1.5% agarose gel electrophoresis and purified using an EasyPure PCR product purification kit (Huayueyang Biotechnology, Beijing, China). The products were sent to Sangon Biotech (Shanghai, China) for sequencing. The target PCR products were amplified and purified as a template for synthesizing dsRNA, following T7 RiboMAX^TM^ Express RNAi System (Promega, Madison, WI, USA) in line with the manufacturer’s instructions. Then the concentration and purity were detected by NanoDrop 2000, and 6 μL dsRNA was detected using 1.5% agarose gel electrophoresis to evaluate integrity. The dsRNA was stored at −80 °C for further RNAi.

#### 2.4.2. Dietary RNAi on Whitefly *Vg* Expression

After fed on either ToCV-infected or uninfected tomato plants for 6 days, approximately 150 whiteflies were carefully transferred into a feeding chamber with 300 μL dsRNA solution (ds*Vg* or ds*GFP*: 300 ng/μL; solution: 15% wt/vol sucrose), respectively [29]. Three replications were conducted for each treatment. The twelve feeding chambers were enveloped by tinfoil, then incubated in the whitefly rearing room for 48 h (Figure 1E). Then RNA was extracted from 30 female whiteflies to detect the expression level of *Vg* after dsRNA treatment. The remaining whiteflies were used for the ovarian development check and fecundity test.

#### 2.4.3. Dietary RNAi on the Development of Whitefly Ovaries

After dsRNA treatment, 15 ToCV-infected and 15 uninfected whiteflies were taken to dissect ovaries. The developmental phases of the oocytes in the ovaries were assessed with a stereomicroscope (Leica, M205C). The maturity of oocytes was determined by the morphological characteristics and the level of yolk content [30,31]. The number of mature oocytes in each treatment was documented.

#### 2.4.4. Dietary RNAi on Whitefly Fecundity

Thirty remaining whiteflies were taken from each tube and then transferred individually into clip-cages on either ToCV-infected or uninfected tomato plants. There was one clip-cage per leaf containing one whitefly. The number of eggs laid by each whitefly was counted with a stereomicroscope after 7 days.

### 2.5. Data Analysis

IBM SPSS Statistics 21 (SPSS Inc., Chicago, IL, USA) was used for statistical analysis. Repeated measures ANOVA was conducted to compare the fecundity and *Vg* expression levels in ToCV-infected and non-infected whiteflies (whiteflies were collected from the same population, with the same development time and individual size). Independent samples *t*-test was also used to compare the silencing efficiency of RNAi, the effect of RNAi on the number of mature oocytes and fecundity in ToCV-infected and uninfected whiteflies. The alpha value (<0.05) was used as a level for judging significance, and a range of alpha from 0.05 to 0.06 was also acceptable for significance.

## 3. Results

### 3.1. Whitefly Fecundity on ToCV-Infected and Uninfected Tomato Plants

During the observation, the number of eggs laid by whitefly increased and reached the peak on 12 d, then decreased gradually (Table 2). Whitefly fecundity was significantly increased on ToCV-infected tomato plants on 12 d (*F* (58) = 9.920, *p* < 0.001), 16 d (*F* (58) = 5.005, *p* =0.012) and 20 d (*F* (58) = 0.310, *p* = 0.002). There was no difference on whitefly fecundity on 4 d (*F* (58) = 0.097, *p* = 0.22), 8 d (*F* (58) = 8.328, *p* = 0.078) and 24 d (*F* (58) = 1.269, *p* = 0.523) (Figure 2A).

### 3.2. Vg Relative Expression Level of Whitefly on ToCV-Infected and Uninfected Tomato Plants

The relative *Vg* expression level was raised while reproduction, reached the peak on 12 d. The expression level started to decline on 16 d, and back to the initial level on 24 d (Table 3). The *Vg* expression level was significantly higher in whiteflies from infected plants than in uninfected plants on 8 d (*F* (4) = 5.699, *p* = 0.057), 12 d (*F* (4) = 3.591, *p* < 0.001) and 16 d (*F* (4) = 4.230, *p* < 0.001), while no significant difference was found in other days (day 4, none *F* and *p* value; day 20, *F* (4) = 3.605, *p* = 0.392; day 24, *F* (4) = 2.478, *p* = 0.075) (Figure 2B).

### 3.3. The Role of Vg on Whitefly Reproduction in the Presence and Absence of ToCV

#### 3.3.1. Vg Expression on Whiteflies with Dietary RNAi

The expression level of *Vg* after feeding with ds*Vg* was significantly decreased compared to ds*GFP*, indicating *Vg* was silenced by 48 h dsRNA oral ingestion. *Vg* expression was decreased by 60.83% on average of whitefly from ToCV-infected plants (SE = 0.011, *t* (4) = 36.327, *p* < 0.001), while 70.75% of whitefly from non-infected plant after RNAi (SE = 0.017, *t* (4) = 62.842, *p* < 0.001) (Figure 3A). The silencing effect of whitefly was better on uninfected plants than on infected ones (*t* (4) = 4.918, *p* = 0.008). This phenomenon could result from *B. tabaci* having higher *Vg* expression from ToCV-infected plants than uninfected before RNAi treatments.

#### 3.3.2. Development of Whitefly Ovaries with Dietary RNAi

ToCV infection increased the number of mature oocytes, while ds*Vg* decreased it. The number of mature oocytes in whitefly on ToCV-infected plants (ds*GFP*: Mean = 5.000, SE = 0.307; ds*Vg*: Mean = 2.000; SE = 0.243; Figure 4A,B) was significantly more than that on uninfected plants (ds*GFP*: Mean = 3.000, SE = 0.330; ds*Vg*: Mean = 1.000; SE = 0.223; Figure 4C,D) (fed with ds*Vg*: *t* (28) = 3.035, *p* = 0.005; ds*GFP*: *t* (28) = 2.810, *p* = 0.009), indicating faster ovary development. Whiteflies fed with ds*Vg* possessed significantly less mature oocytes than ds*GFP* treatments no matter on plants infected with ToCV or not (uninfected: *t* (28) = 5.187, *p* < 0.001; infected: *t* (28) = 5.965, *p* < 0.001) (Figure 3B).

#### 3.3.3. Whitefly Fecundity with Dietary RNAi

Whitefly fecundity was increased by ToCV infection, but decreased by ds*Vg*. In non-viruliferous whitefly, the number of eggs after fed with ds*Vg* (mean = 24.000, SE = 0.793) was significantly lower than the egg number with ds*GFP* (mean = 40.000, SE = 0.584) (*t* (58) = 15.941, *p* < 0.001). In viruliferous whitefly, the egg number after ds*Vg* treatments (mean = 28.000, SE = 0.839) was also significantly lower than with ds*GFP* (mean = 45.000, SE = 0.616) (*t* (58) = 16.623, *p* < 0.001). The number of eggs laid by viruliferous whitefly was significantly higher than that non-viruliferous whitefly (fed with ds*Vg*: *t* (58) = 3.119, *p* = 0.003; ds*GFP*: *t* (58) = 6.125, *p* < 0.001) (Figure 3C).

## 4. Discussion

### 4.1. ToCV Infection Facilitated the Whitefly Reproduction

The spread of a vector-borne plant pathogenic virus mainly relies on the performance and behavior of the vectors, usually insects. Such viruses should be expected to be adaptive to affect the physiological or behavioral characteristics of their insect vectors either direct through modifying the performance and behavior of the vectors or indirect through manipulating their shared host plants to make them more attractive or acceptable to their vectors to improve the viruses transmission and benefit in viruses fitness [8].

In whiteflies, such effects could be different among different biotypes or infected by different viruses, requiring investigations case by case (reviewed by Eigenbrode et al., 2018, supplemental Table 1). The reproduction has been increased in another biotype whitefly (biotype B) when infected by tobacco curly shoot virus (TbCSV) or tomato yellow leaf curl China virus (TYLCCNV) [32], so as an acceleration of ovarian development on TYLCCNV [33]. However, when infected with ToCV, the *B. tabaci* MEAM1 shows no preference between the ToCV-infected plant and uninfected plants, and the oviposition and survival rate are decreased. This results can explain why B. tabaci MED has replaced MEAM1 in the past decade in China. However, ToCV infection did not influence the number of eggs laid by *Trialeurodes vaporariorum* [16,34,35]. Previous research confirmed the *B. tabaci* MED fecundity affected by the semi-persistent virus, ToCV [16,36,37], which is consistent with our observation. The ToCV-infected tomato plants could accelerate *B. tabaci* MED ovarian development and enhance their fecundity to increase the vector abundance, and then promote the virus spread. Our findings confirm the key role of the *B. tabaci* MED in the spread of ToCV, indicating the importance of whitefly control in the control of ToCV [16]. The underlying mechanisms are actually important to develop new tools that may be included in an integrated pest management program to control whiteflies.

### 4.2. ToCV Infection Facilitates Whitefly Reproduction through Elevating Vg Expression

Studies about the molecular mechanisms of these effects of viruses modulating their vectors to facilitate their transmission are still rare. In our study, ToCV-infection increased the *Vg* expression, along with the fecundity. Both ovarian development and fecundity were suppressed when *Vg* expression was knocked down, indicating that the increase of whitefly fecundity on ToCV-infected plants is consistent with the Vg gene. This correlation could be either direct or indirect by the virus. The *Vg* expression could be directly increased by the acquisition of the virus, or indirectly, the virus might mediate the plant quality first. Nutrient change in the host plants could then induce the *Vg* expression, facilitating the whitefly reproduction. It has been demonstrated that TYLCV could benefit whitefly fecundity by altering the nutritional content of the host’s leaf tissue and phloem sap [38,39]. Evidence about these plant regulations to enhance vector performance has also been found in other insect vector-virus systems, such as Barley yellow dwarf virus and Cucumber mosaic virus with aphids [40,41], Tomato spotted wilt virus with thrips [42].

There was no evidence of direct interaction between ToCV and vitellogenin before. ToCV is a semi-persistent and foregut-borne virus, and interestingly, the whiteflies fixed on ToCV-infected plants showed the induced fecundity and vitellogenin gene expression. The host modulate may be the main reason. The limitation of this research was that we did not evaluate host nutrient status with and without ToCV infection, which need further investigation. A previous transcriptome study about ToCV-infected whitefly found another vitellogenin related gene, *Vitellogenin-B* (*Bta15563*) as one of the most down-regulated genes after 24 h [20]. The effect of ToCV on the expression of vitellogenin related genes might be different along with the infectious stages. In the meantime, other differential expressed genes were also been identified in this previous study, suggesting other potential genes associated with whitefly vitellogenin or reproduction-related pathways during ToCV infection.

### 4.3. RNAi on Vg as a New Approach in ToCV/Whitefly Control

Understanding the chemical and molecular mechanisms of such effects as viruses influencing vectors has significant importance for evolutionary perspective on viruses’ adaptation to this complex interaction and applications, providing opportunities to improve our current pest-control method or finding new ones. In our study, the dietary ds*Vg* could effectively suppress the expression level of *Vg* in both ToCV-infected and uninfected whitefly, reducing the fecundity of whiteflies. The population of whitefly is expected to be suppressed. As ToCV cannot be transmitted to the host plants in the absence of *B. tabaci* vectors, the transmission of virus should then be constrained. Chemical control of the vector has not been demonstrated to be effective for controlling the disease caused by ToCV [19]. RNAi technology has been an effective tool in functional genomics studies and its application toward pest management is already close to reality [43]. Furthermore, RNAi will be safer than any pesticide currently available in the registration because of the high specificity, typically targeting a single gene [44]. The field of dietary RNAi is relatively new, and many questions regarding the fate of dsRNA in the environment, effects on non-target organisms, risk prediction, etc. still remain explored at this moment. The costs, efficiency, benefits and risks of pesticidal RNAi must be considered while it moves from a theoretical approach to being used as a practical tool [45].

## 5. Summary and Perspectives

In summary, this study shows that ToCV-infected tomato plants increases *B. tabaci* fecundity by elevating *Vg* expression, providing an underlying molecular mechanism for the fecundity of vector modulated by the plant pathogenic virus to facilitate their own spread. A future study focusing on other differentially expressed genes during the ToCV infection is warranted. Given that whitefly fecundity was significantly suppressed by dietary RNAi, future research on new molecular target(s) could fundamentally reduce whitefly abundance.

## Figures and Tables

**Figure 1 insects-12-00101-f001:**
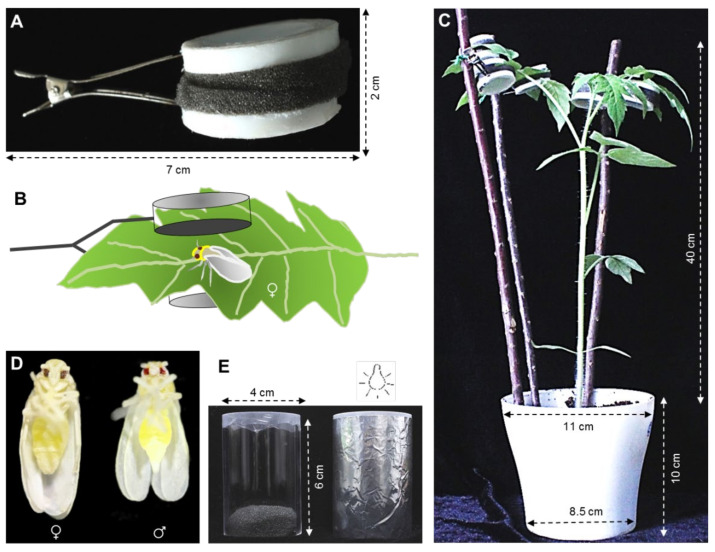
Experimental setup. (**A**) Clip-cage; (**B**) A female whitefly was contained in one clip-cage attached to one tomato abaxial leaf; (**C**) Five clip-cages in which a female whitefly respectively attached to tomato plant; (**D**) Female and male whiteflies; (**E**) Feeding chamber, dsRNA is contained between two layers of parafilm in the end of glass feeding tube. After whiteflies are released into the other end of the tube, the tube is sealed with a black cotton plug, covered with tinfoil, and oriented with the parafilm-covered end toward a light source about 0.5 m away in whitefly rearing room for feeding 48 h.

**Figure 2 insects-12-00101-f002:**
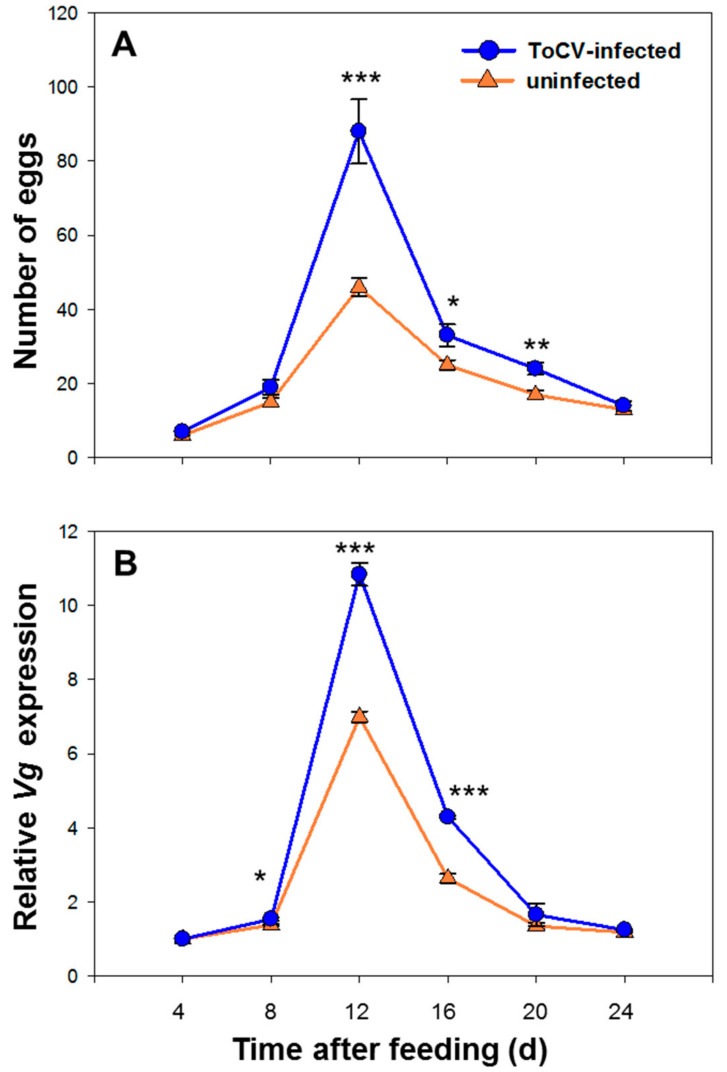
Temporal change of whitefly fecundity and *Vg* expression on ToCV-infected and uninfected tomato plants. (**A**) represents the dynamic change of egg numbers within 24 days (Mean ± SE, n = 30); (**B**) represents the dynamic change of relative *Vg* expression within 24 days (Mean ± SE, n = 30). The asterisks above bars mean significant difference: * *p* ≤ 0.05, ** *p* ≤ 0.01, *** *p* ≤ 0.001. Data were pooled with 3 replications.

**Figure 3 insects-12-00101-f003:**
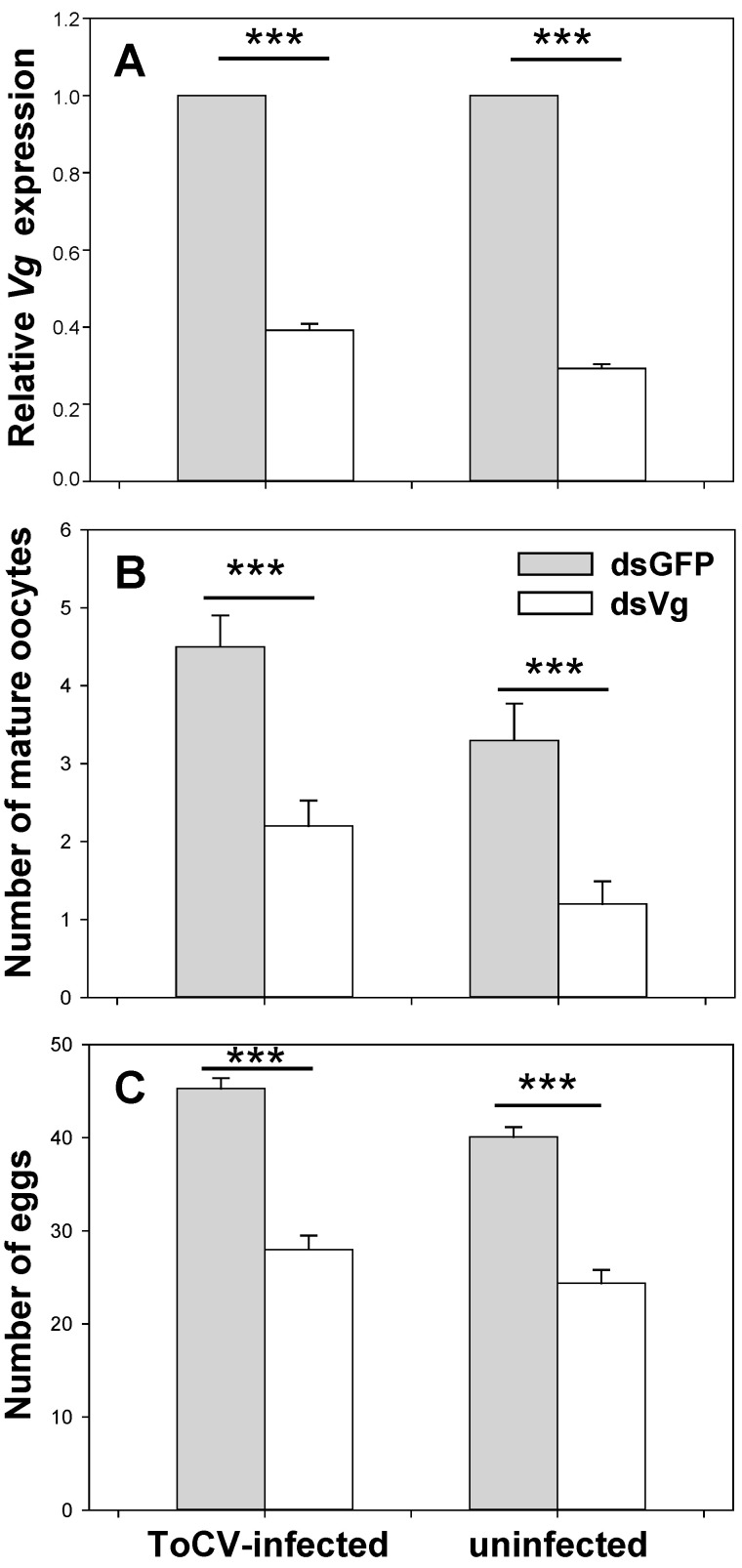
The role of *Vg* on whitefly reproduction in the presence and absence of ToCV. In dietary RNAi, *B. tabaci* MED was fed with ds*GFP* and ds*Vg* on ToCV-infected and uninfected tomato plants. After 48 h, relative *Vg* expression (Mean ± SE, n = 30, **A**), the number of mature oocytes (Mean ± SE, n = 15, **B**), and eggs (Mean ± SE, n = 30, **C**) were documented. The asterisks above bars mean significant differences (*** *p* ≤ 0.001). Data were pooled from 3 biological replications.

**Figure 4 insects-12-00101-f004:**
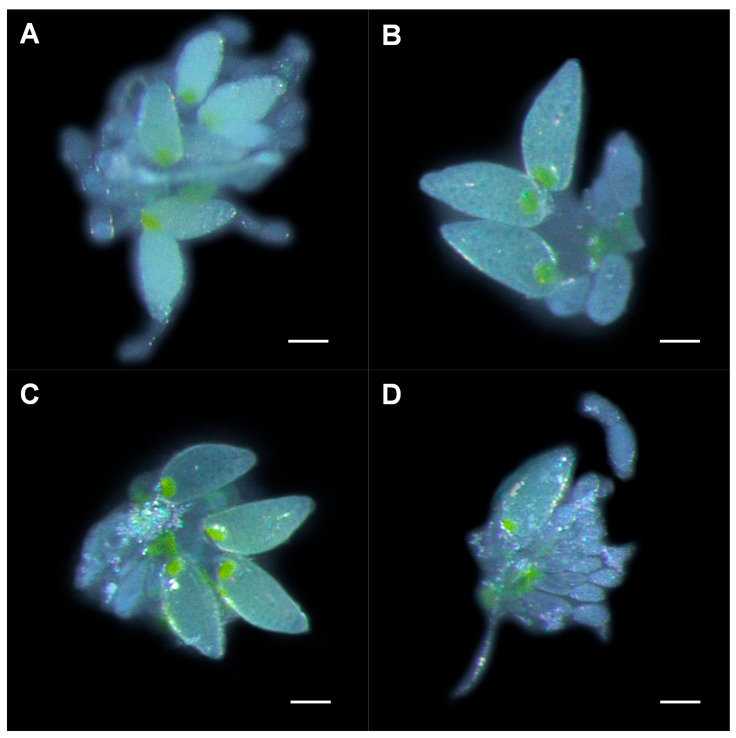
Ovary development in whiteflies that fed on ToCV-infected and uninfected plants post-dietary RNAi. Ovaries from (**A**) viruliferous whitefly after ds*GFP* ingestion, (**B**) viruliferous whitefly after ds*Vg* ingestion, (**C**) non-viruliferous whitefly after ds*GFP* ingestion, and (**D**) non-viruliferous whitefly after ds*Vg* ingestion. Scale bar: 0.10 mm.

**Table 1 insects-12-00101-t001:** Primers used in this study.

Gene	Primer Sequence (5′-3′) *
*Actin*	F: CGCTGCCTCCACCTCATT R: ACCGCAAGATTCCATACCC
*EF-1α*	F: TAGCCTTGTGCCAATTTCCG R: CCTTCAGCATTACCGTCC
qPCR *Vg*	F: CTTCTCCGCTGCTTTCTT R: TGTTGGCGTATTTGTTGG
ds*GFP*	F: TAATACGACTCACTATAGGGTTCAGTGGAGAGGGTGAAGGT R: TAATACGACTCACTATAGGGTGTGTGGACAGGTAATGGTTG
ds*Vg*	F: ATTCTCTAGAAGCTTAATACGACTCACTATAGGGTCTGTGATGCCTTAGTT R: ATTCTCTAGAAGCTTAATACGACTCACTATAGGGCTCTCTTGAGGTTTTGT

* F denotes forward primer; R denotes reverse primer.

**Table 2 insects-12-00101-t002:** Number of eggs laid by whitefly on tomato plants.

Treatments	Time					
	4 d	8 d	12 d	16 d	20 d	24 d
ToCV-infected	7.0 ± 8.668	19.0 ± 1.988	88.0 ± 8.668	33.0 ± 3.016	24.0 ± 1.627	14.0 ± 1.314
Uninfected	6.0 ± 0.931	15.0 ± 1.192	46.0 ± 2.441	25.0 ± 1.315	17.0 ± 1.181	13.0 ± 0.965

Note: the values are mean ± SE.

**Table 3 insects-12-00101-t003:** *Vg* relative expression level of whitefly.

Treatments	Time					
	4 d	8 d	12 d	16 d	20 d	24 d
ToCV-infected	1.000 ± 0.000	1.541 ± 0.093	10.840 ± 0.537	4.290 ± 0.037	1.656 ± 0.303	1.248 ± 0.002
Uninfected	1.000 ± 0.000	1.380 ± 0.030	6.995 ± 0.253	2.640 ± 0.117	1.353 ± 0.089	1.184 ± 0.003

Note: the values are mean ± SE.

## Data Availability

Not applicable.

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
