# Peer review of "Tomato Chlorosis Virus Infection Facilitates Bemisia tabaci MED Reproduction by Elevating Vitellogenin Expression"

_insects, 2021, doi:10.3390/insects12020101_

Round 1

Reviewer 1 Report

The study shows that B. tabaci MED species feeding on ToCV-infected tomato plants had an increase in the fecundity by elevating vitellogenin expression compared to a population feeding on uninfected tomato plants. The methods and results are very interesting and I recommend the manuscript for publication on Insects. However, there are some sections of the manuscript that needs major improvements and some points to be corrected and clarified by the authors.   

Title: Bemisia tabaci is a cryptic species and the species used in the experiment (MED or Q biotype) must be included in the title

L20: Replace “corps” by crops

L22: Tomato chlorosis virus and the other virus names throughout the manuscript should not be italicized.

L25: Replace “slienced” by silenced

Introduction

This section is very short and needs improvement. The paragraph introducing the whitefly (L:64-75) should be placed before the paragraph introducing the role of vitellogenin. In addition, the authors should know that MEAM1 and MED, or can be also named as biotypes, are different species and have completely different characteristics. Therefore, every time a previous study using either of the species of the complex B. tabaci  is mentioned, it needs to be clarified which species was used.

L55: Whenever the species is first mentioned in the manuscript, is important to refer to the other names: biotype Q also known as Mediterranean or MED

L58: “whitefly” here refers to B. tabaci MEAM1 (or B biotype) species

Methods

L91: Shorten Bemisia tabaci: B. tabaci

L97: “Attached to one leaf” is repeated twice

L102: replace “was” by “were”

L117: Why transferring whiteflies to new plants every 12 days? Adults that were previously on infected plants were transferred to infected plants, and adults on uninfected were transferred to uninfected plants? Why did you do that? The plants were getting too old? Please clarify and add to the manuscript.

L120: What method of RNA extraction was used? The RNA was extracted from a pooled sample of 30 individuals? Please clarify.

L127 and L129: You mentioned that three technical replications were used twice in the text.

Results

L203-204: The same sentence repeated twice. Please correct.

L206: There is any hypothesis why the silencing was more efficient on uninfected whiteflies compared to infected whiteflies?

L232: Missing a period at the end of the sentence, after the parenthesis.

L233: replace by “eggs laid”; and replace by “significantly higher”

Discussion

L246: virus names are not italicized here.

L249: In addition to these references, you should mention and compare your results with the study by Watanabe et al., 2018 (DOI: 10.1111/jen.12559) which shows the effects of ToCV on the performance of two other whitefly species, the B. tabaci MEAM1 and Trialeurodes vaporariorum.

L254: The last sentence should be rephrased. The underlying mechanisms are actually important to develop new tools that may be included in an integrated pest management program to control whiteflies.

L257: “studies are rear”? Did you mean “rare”?

L270: Include the information that this study was carried out with MEAM1 species. This information (down-regulation of Vitellogenin MEAM1) is very interesting as the effects on MEAM1 feeding on ToCV-infected tomatoes are negative for the vector (see Watanabe et al.,  J Appl Entomol. 2018;1–8). In contrast, MED has positive effects on the vector behavior and up-regulation of Vitellogenin. I believe this should be discussed in the manuscript.

L273: “In the meantime” is more appropriate

L289: Again, you must be careful with this kind of sentence. I don’t agree that this is a new method to control ToCV transmission by reducing the oviposition of whiteflies. This is only an in vitro study and certainly has limitations when applied to the field. In addition, reducing the whitefly oviposition doesn’t mean controlling the virus, because one whitefly individual is enough to transmit the virus, it’s very common to find tomato field heavily affected by ToCV with low whitefly infestations. Also, the experiment was carried out with B. tabaci MED species and may not be applied for the other major invasive whitefly species, the MEAM1. The results obtained are very interesting but it’s only a very first step until it becomes a ”new method to control ToCV transmission and whiteflies”.

Author Response

Manuscript insects-1023077 entitled “Tomato chlorosis virus infection facilitates Bemisia tabaci reproduction by elevating vitellogenin expression” has been revised according to the editorial suggestions. Reviewers’ constructive criticisms and suggestions are greatly appreciated. We have essentially followed most of the editorial changes suggested by the reviewers, and the remaining comments and suggestions have been incorporated into the revised manuscript (highlighted in yellow). The following is a point-to-point response to reviewers’ comments.

Reviewer 2 Report

The article is interesting, provides valuable results and should be published after some necessary modifications and additions to improve clarity.  The entire paper should also be copy-edited for needed English grammar and composition corrections.

1. A key question is how viruliferous status was determined in whiteflies?  Whiteflies were given infected and non-infected host plant material, but was virus infection determined in the whiteflies themselves?  It does not appear that this was done, and should be clarified.  

2. The key experiment I looked for was a comparison of vitellogenin and ovarian activity in infected vs. non-infected whiteflies, but did not see it. This is probably a result of design given in # 1 above.

line 111.  Figure 1 B.  The feeding chamber is hard to visualize and a clear diagram together with the photograph would be helpful.  It's also not clear what exactly is meant by "dsRNA is contained between layers of parafilm".

line 169.  Repeated measures ANOVA was used and is appropriate for some of the variables reported, and in other cases t-tests were done.  However the methods description does not explicitly state that certain variables were measured over time from the same samples, thereby qualifying for repeated measures, which should be clarified with a simple sentence.

Also in the statistical methods, the critical level of alpha (Type 1 Error) adopted for judging significance should be specified.  The authors implicitly use 0.05 as a critical level but do not justify it.  In this sort of experimentation, a range of alpha from 0.05 to 0.10 is very acceptable but is not used here.  For example, in judging differences in Vg Relative Expression on Day 8, results are given as "not significant" with a P (alpha) of 0.057.  This is a flawed interpretation of P values, and this difference should be qualified to be significant at the 0.057 level, unless justified otherwise.

Results given in sections 3.1 and 3.2 are very hard to read and would be much better presented in simple tables.  This is a NECESSARY change.

line 252.  Comment should be given on merits of further investigating preference for infected host plants to determine survival and reproductive potential (whitefly fitness due to infection).

line 283.  It is interesting and intriguing to speculate on the management implications, but some comment on how dietary dsVg might be delivered to a field environment should be included.

Author Response

(The authors gave the same response as above.)

Reviewer 3 Report

The review comment is in attachment.

Author Response

(The authors gave the same response as above.)

Reviewer 4 Report

The manuscript by Huang et al “Tomato chlorosis virus infection facilitates Bemisia tabaci reproduction by elevating vitellogenin expression” makes an interesting connection between vitellogenin expression and whitefly fitness following ToCV infection. However, the manuscript makes a misleading leap to suggest that ToCV infection itself could be directly affecting the fitness responses on the whiteflies, this manuscript establishes a correlation but does not provide evidence for causation.

Manuscript in numerous instances suggests that the ToCV infection facilitates these vitellogenin guided fitness effects on whiteflies. What the manuscript fails to explain early on is that ToCV is a semi-persistently transmitted virus. Semi-persistent viruses are foregut borne and could be lost relatively quickly. The question then becomes how would a foregut-borne virus be directly associated with vitellogenin production? I am not disputing the link between vitellogenin gene expression/vitellogenin production with enhanced fecundity and/or whitefly fitness. In fact, that link has long been established. It is also well established that host nutrient status/food source nutrient status could influence vitellogenin production. The ToCV infection could be responsible for enhancing vitellogenin gene upregulation, but that does not mean ToCV is directly responsible for this upregulation. Rather, it is very likely that the modulation/increase in phloem nutrients could be causing this spike in vitellogenin upregulation in insects. Considering the mode of transmission, unless the authors establish a direct link between ToCV and Vitellogenin upregulation, it is safe to assume that such effects on whitefly fitness are modulated via the plant host. This aspect is very critical, and it is misrepresented in the manuscript. The manuscript requires major rewriting to incorporate the above-stated suggestion/criticism.

Besides, the replications in a few instances are very few and the marginal fitness effects obtained using these experiments warrant further scrutiny (details to follow). As a minor note, the manuscript is also riddled with grammatical errors so much so that it needs a thorough editing. I attempted language editing but had to quickly give up. Other minor issues include improperly italicizing virus species names and not including author names for binomial names.

As written, the manuscript is misleading. It would take substantial major revisions to rewrite this manuscript within the framework of experiments conducted and results obtained. The authors would also have to acknowledge the limitations (not evaluating host nutrient status with and without ToCV infection is a primary one).  The minor changes and details are highlighted below.

Line 25: ‘Manipulates’- Manipulates is very anthropomorphic. I suggest using modulate instead throughout. 

Lines 34-35: I do have an issue with the hypothesis. Why would a forgut-borne semi-persistently transmitted virus influence gene expression in the ovaries?  This abstract and paper fail to distinguish direct virus-mediated effects from indirect host-mediated effects. Without that this hypothesis is flawed to begin with. 

There is no dispute that vitellogenin is essential for whitefly reproduction and fecundity. The issue here is that ToCV may not be directly modulating vitellogenin expression, but rather indirectly altering the nutrient profile of the host plant upon infection.

Lines 49-50: Viruses are the causal agents of diseases. Therefore, viruses threaten crop production/agriculture

Lines 49-51. Please use plural form for nouns including vectors and viruses.

Line 53: There is confusion between ‘virus’ and ‘disease’ usage. Virus is the causal agent of the disease. Disease is a phenotype resulting from virus-host interactions.

Lines 56-57. Please edit for grammar

Lines 59-60. Please follow ICTV conventions for writing virus names. Only taxonomic references of the virus should be italicized. Physical form of the virus should not be.   

Line 68: millions to billions of what? Please specify

Line 73. Please rephrase.

Lines 75-77: This is a comparative statement. So, compared with what?

Lines 81-83: Goes back to my original criticism.

Line 90: Please include author names for all binomial names.

Figure 1: Please rephrase ‘by using a stereo…

Line 123: 200 microliters from how many insects? Please specify.

Line 132: Very minimal number of biological (3?) replications.

Line 164: Should be ‘30 remaining’

Line 177: ‘facilitated by ToCV’ Not true. There were more eggs on ToCV-infected plants. But it is not known if this is a direct effect modulated by ToCV or an indirect effect modulated via the host plant.

Results overall: Please include numerator and denominator degrees of freedom for all F values.

I also disagree with the usage of ‘infected and uninfected’ whiteflies. Is there evidence for virus infection in whiteflies? I would rather refer to them as viruliferous and non-viruliferous whiteflies. Again, keep in perspective that this is a semi-persistently transmitted virus.

Lines 209-210. What does this refer to vg expression? Explanation for figs. 3B and 3C?

Lines 240-241: This sentence needs to be rephrased. This depends upon the type of virus and mode of transmission. Non-persistent viruses transmitted by non-colonizing vectors do not fit this profile. Generic statements should be accompanied by plural noun forms. 

Lines 259-260: Please clarify. Rare? Not ‘rear’.

Lines 263-265: This is the most important point, and it has to be emphasized throughout the manuscript.

Lines 271-278: Still, no evidence of direct interaction between ToCV and Vitellogenin. The question remains how would foregut-borne virus directly impact ovaries? No premise and no evidence. This should be predominantly host modulated.

Where in the vitellogenin pathway do the authors believe ToCV could be involved? Nutrients are the drivers of vitellogenin production isn’t it? That part seems to be completely discounted here. Unless that is identified, there is no evidence of ToCV directly regulating vitellogenin production.

Lines 295-296: Misleading. Indirect host-modulated changes are critical to understand first.

Author Response

(The authors gave the same response as above.)
